# Quadruple bonding between iron and boron in the BFe(CO)$_3$$^-$ complex

Chaoxian Chi [1,5], Jia-Qi Wang[2,5], Han-Shi Hu[2]*, Yang-Yang Zhang[2], Wan-Lu Li[2], Luyan Meng[1], Mingbiao Luo[1], Mingfei Zhou [3]* & Jun Li[2,4]*

While main group elements have four valence orbitals accessible for bonding, quadruple bonding to main group elements is extremely rare. Here we report that main group element boron is able to form quadruple bonding interactions with iron in the BFe(CO)$_3$$^-$ anion complex, which has been revealed by quantum chemical investigation and identified by mass-selected infrared photodissociation spectroscopy in the gas phase. The complex is characterized to have a B-Fe(CO)$_3$$^-$ structure of C$_{3v}$ symmetry and features a B-Fe bond distance that is much shorter than that expected for a triple bond. Various chemical bonding analyses indicate that the complex involves unprecedented B≡Fe quadruple bonding interactions. Besides the common one electron-sharing σ bond and two Fe→B dative π bonds, there is an additional weak B→Fe dative σ bonding interaction. This finding of the new quadruple bonding indicates that there might exist a wide range of boron-metal complexes that contain such high multiplicity of chemical bonds.

[1] School of Chemistry, Biological and Materials Sciences, East China University of Technology, 330013 Nanchang, Jiangxi Province, China. [2] Department of Chemistry & Key Laboratory of Organic Optoelectronics and Molecular Engineering of Ministry of Education, Tsinghua University, 100084 Beijing, China. [3] Department of Chemistry, Shanghai Key Laboratory of Molecular Catalysis and Innovative Materials, Fudan University, 200433 Shanghai, China. [4] Department of Chemistry, Southern University of Science and Technology, 518055 Shenzhen, China. [5]These authors contributed equally: Chaoxian Chi, Jia-Qi Wang. *email: hshu@mail.tsinghua.edu.cn; mfzhou@fudan.edu.cn; junli@tsinghua.edu.cn

Chemical bonding is among the most fundamental concepts in modern chemistry. Depending upon the availability of valence orbitals and electrons, an atom can form single or multiple covalent chemical bonds with neighboring atoms in forming stable compounds. Since the discovery of Re–Re quadruple bond in 1964 by Cotton and coworkers[1,2], metal–metal multiple bonding with bond orders of four to six has been extensively explored for transition metals and actinides[3–18]. Main group elements of the periodic table may form single, double, and triple bonds between two atoms, that is, the maximum bond order can only be three as exemplified in alkyne. Triple bonds between two main group atoms are well established, and a large number of triple-bonded molecules are known[19–26]. As main group elements have $nsnp$ four valence orbitals accessible for bonding, it has been speculated that a further extension to bond order of four for main group elements should in principle be possible. Diatomic $C_2$ and its isoelectronic molecules $CB^-$, BN, and $CN^+$ each having eight valence electrons were claimed to be quadruple-bonded molecules, comprising not only one σ- and two π-bonds, but also one weak 'inverted' bond[27]. However, this quadruple bond assignment has been debated, which has stimulated hot discussion[28–30]. Quadruple bonding of carbon to uranium with rich valence shell (sdf) has been reported to exist in the triatomic uranium carbide oxide molecule CUO and related species, due to availability of both 2p and unhybridized 2s orbitals of carbon[31,32]. Besides the ubiquitous one σ- and two π-bonds, a non-negligible, albeit weak, rearward σ-bond was characterized to exist between C and U atoms.

As an electron-deficient atom, boron prefers to form delocalized multicenter bonds and resists multiple bonding in general. However, boron is capable of forming homoatomic B-B double and triple bonds as well as multiple bonds to other elements. In 2002, one of us reported the first B≡B triple bond in (OC)BB(CO) formed in a low temperature argon matrix[20]. This work was followed by Wang and coworkers on the gas-phase characterization of the isoelectronic, B≡B triple-bonded dianionic molecule [(OB)BB(BO)]$^{2-}$ [21]. These reports were accompanied by a flurry of theoretical studies exploring a range of donors for stabilizing the triple-bonded $B_2$ unit[33–35]. The first chemical compound containing a B≡B triple bond that is stable at ambient temperature was synthesized in 2012 by Braunschweig and coworkers and was confirmed by X-ray crystallography[22,23]. Boron can also form multiple bonds with metal atoms, and a number of complexes featuring transition metal-boron double bond have been characterized[36–38]. However, boron-metal multiple bonding with bond orders above two is elusive, although boron-metal triple bonds were found in nominal MB diatomics[39,40]. Only a bismuth-boron cluster anion, BiB(BO)$^-$ was formed in the gas phase, which was characterized by photoelectron spectroscopy and theoretical calculations to contain a bismuth-boron triple bond[41]. Complexes featuring boron-transition metal quadruple bonding interactions are currently unidentified.

Here we report that main group element boron is able to form quadruple bonding interactions with transition metal iron in the BFe(CO)$_3^-$ anion complex, which is found through advanced quantum chemical calculations and is experimentally generated in the gas phase and characterized by mass-selected infrared photodissociation spectroscopy.

## Results

**Mass spectra.** The iron-boron heteronuclear carbonyl anion complexes were generated in the gas phase using a pulsed laser vaporization/supersonic-expansion ion source[42]. The anion complex was mass-selected by its flight time and was studied by

infrared photodissociation spectroscopy in the carbonyl stretching vibrational frequency region (see Methods). The mass spectrum in the $m/z$ range of 100–250 obtained by pulsed laser vaporization of a boron-10 depleted target in expansions of helium gas seeded with 7% CO is shown in Fig. 1. The spectrum is dominated by the peak at $m/z = 168$ that can be assigned to the Fe(CO)$_4^-$ anion, which is a common species observed in the experiments using the carbon monoxide samples containing trace of iron carbonyl impurity. Besides the most intense Fe(CO)$_4^-$ anion peak, the next most intense peaks are observed at $m/z = 151$ and $m/z = 207$, which can be assigned to the $^{11}$BFe(CO)$_3^-$ and $^{11}$BFe(CO)$_5^-$ anions, respectively. Similar experiments with natural abundance boron and boron-10 enriched targets were also performed, and the mass spectra are shown in Supplementary Fig. 1. Additional experiments using a $^{13}$C-substituted CO sample were performed to confirm that the observed species at $m/z = 151$ is due to $^{11}$BFe(CO)$_3^-$ rather than the equal mass $^{11}$B(CO)$_5^-$ or $^{11}$BFe$_2$CO. A mixed target involving boron-10 depleted boron and iron was used to incorporate iron atoms into the system, as the $^{13}$C-substituted CO sample does not involve any iron carbonyl impurity. These experiments provide conclusive identification of the BFe(CO)$_3^-$ anion complexes.

**Infrared photodissociation spectra.** The BFe(CO)$_3^-$ anion was selected for infrared photodissociation by a tunable IR laser beam. The anion photo-dissociates via the loss of one CO ligand when the IR laser is on resonance with the carbonyl stretching fundamentals of the anion. The dissociation is observed only under focused infrared light irradiation with very low efficiency. The parent anions were depleted by less than 3% at the laser pulse energy of ~0.8 mJ per pulse at 1841 cm$^{-1}$. Such low dissociation efficiency implies that the BFe(CO)$_3^-$ anion is a strongly bonded species and the dissociation is due to a multiphoton process. The IR photodissociation spectrum of $^{11}$BFe(CO)$_3^-$ obtained by monitoring the fragment ion yield as a function of the dissociation IR laser wavelength in the carbonyl stretching frequency region is shown in Fig. 2. The spectrum contains only two bands centered at 1841 and 1911 cm$^{-1}$. The observation of only two carbonyl stretching bands suggests that the anion complex should have C$_{3v}$ symmetry with all the three carbonyl ligands bonded to the same center. The bands are quite broad due to power broadening and/or the involvement of hot anions, as the

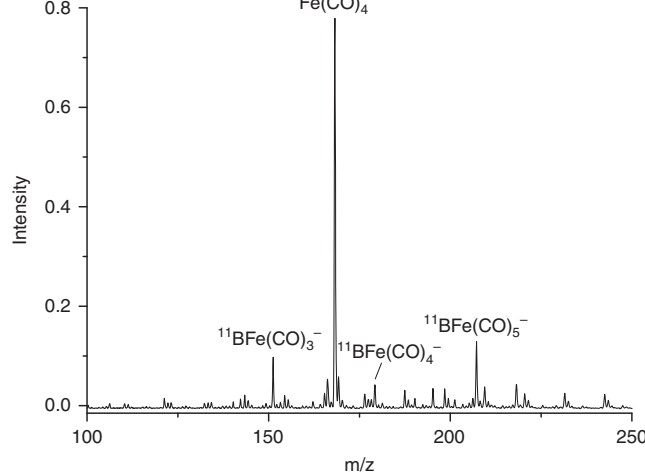

**Fig. 1** Mass spectrum of the boron-iron carbonyl anion complexes. The complexes are formed by pulsed laser vaporization of a boron-10 depleted target in an expansion of helium seeded by 7% carbon monoxide with trace of iron carbonyl impurity

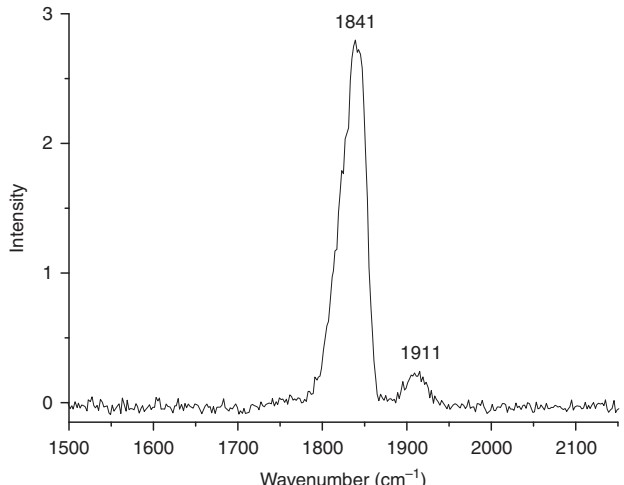

**Fig. 2** Infrared photodissociation spectrum of the $^{11}BFe(CO)_3^-$ anion complex. The spectrum is measured in the carbonyl stretching frequency region by monitoring the CO photodissociation channel. Intensity is shown as the yield of fragmentation ion normalized to the parent ion signal in percentage

**Table 1 Calculated carbonyl stretching frequencies (in cm$^{-1}$) and intensities (in parentheses, in km mol$^{-1}$) as well as the experimental vibrational frequencies of the BFe(CO)$_3^-$ complex**

| | sym. str. | | asym. str. | |
|---|---|---|---|---|
| | $^{11}$BFe ($^{12}$CO)$_3^-$ | $^{11}$BFe ($^{13}$CO)$_3^-$ | $^{11}$BFe ($^{12}$CO)$_3^-$ | $^{11}$BFe ($^{13}$CO)$_3^-$ |
| PBE | 1929 (227) | 1881 (219) | 1860 (1384) | 1816 (1313) |
| B3LYP[a] | 1935 (308) | 1889 (289) | 1859 (1748) | 1817 (1662) |
| M06-2X[a] | 1966 (292) | 1918 (276) | 1847 (2177) | 1805 (2066) |
| Exptl. | 1911 | 1859 | 1841 | 1797 |

[a]The scale factors for B3LYP and M06-2X are 0.97 and 0.94, respectively, taken from the ratio of the calculated harmonic frequency and the experimental frequency (2143 cm$^{-1}$) for free CO as shown in Supplementary Table 6

dissociation is a multiphoton process. The spectra of the $^{10}BFe(CO)_3^-$ and $^{11}BFe(^{13}CO)_3^-$ isotopomers are also recorded as shown in Supplementary Fig. 2. The $^{11}BFe(^{13}CO)_3^-$ spectrum shows two bands at 1797 and 1859 cm$^{-1}$. The isotopic shifts are appropriate for carbonyl stretching vibrations. The band positions of $^{10}BFe(CO)_3^-$ are only slightly blue-shifted from those of $^{11}BFe(CO)_3^-$, indicating that the two observed modes are pure carbonyl stretching vibrations with negligible involvement of the boron atom. This observation also indicates that the three carbonyl ligands are coordinated on the iron center. Accordingly, the experimentally observed BFe(CO)$_3^-$ anion should have a B–Fe bonded B–Fe(CO)$_3^-$ structure with $C_{3v}$ symmetry. The IR photodissociation spectra of the Fe(CO)$_4^-$, BFe(CO)$_4^-$, and BFe(CO)$_5^-$ anions are also recorded as shown in Supplementary Figs. 3–5. The IR spectrum of BFe(CO)$_3^-$ is quite different to those of the Fe(CO)$_4^-$, BFe(CO)$_4^-$ and BFe(CO)$_5^-$ anions, implying different structures and bonding patterns in these anion complexes.

**Theoretical results**. Quantum chemical calculations with density functional theory (DFT) and wavefunction theory (WFT) (see Methods) on the BFe(CO)$_3^-$ complex and the other species observed reveal that the BFe(CO)$_3^-$ complex is rather stable and has quadruple bonding interactions between boron and iron. The B–Fe bonded $C_{3v}$ structure with all the carbonyl ligands coordinated to the Fe center was predicted to have a closed-shell singlet ground state as shown in Supplementary Table 1. The other isomers with one or two CO ligands bonded to the boron center, (OC)BFe(CO)$_2^-$ and (OC)$_2$BFe(CO)$^-$ were predicted to have triplet electronic ground states. The high-level ab initio DLPNO-CCSD(T) calculations predicted that the $C_{3v}$ BFe(CO)$_3^-$ isomer is more stable than the (OC)$_2$BFe(CO)$^-$ and (OC)BFe(CO)$_2^-$ structures by 27.8 and 42.5 kcal mol$^{-1}$, respectively. The calculated infrared spectra of the BFe(CO)$_3^-$, (OC)BFe(CO)$_2^-$, and (OC)$_2$BFe(CO)$^-$ structures are compared with the experimental spectrum in Supplementary Fig. 6. The simulated spectrum of the most stable $C_{3v}$ BFe(CO)$_3^-$ structure is consistent with the experimental spectrum. Although the main experimental band is quite broad with a full width at half-maximum (fwhm) of ~35 cm$^{-1}$, the high-lying (OC)$_2$BFe(CO)$^-$ and (OC)BFe(CO)$_2^-$ isomers are unlikely presented in the experiment because they are

energetically much higher than the $C_{3v}$ BFe(CO)$_3^-$ structure. Table 1 lists the calculated CO stretching frequencies and IR intensities of the most stable structure together with the experimental values. The agreement between theory and experiment supports the geometric and electronic structure assignments of the observed BFe(CO)$_3^-$ anion complex. Unfortunately, the Fe–B stretching frequency of BFe(CO)$_3^-$ cannot be directly observed as it was predicted at the B3LYP level to absorb at 863 cm$^{-1}$ with very low IR intensity (5 km mol$^{-1}$).

Geometry optimizations with DFT functionals give rise to B–Fe distances of 1.69, 1.67, and 1.61 Å, respectively, at the PBE, B3LYP, and M06-2X levels. DLPNO-CCSD(T) calculation provides an equilibrium B–Fe bond distance of 1.63 Å (Supplementary Fig. 7). These values are considerably shorter than the sum of the triple-bond covalent radii of iron and boron atoms proposed by Pyykkö and coworkers (B + Fe = 1.75 Å)[43]. Such a short bond distance suggests that the B–Fe bond in BFe(CO)$_3^-$ should have a bond order higher than three. The calculated bonding dissociation energy ($D_e$) is 124.6 kcal mol$^{-1}$ at the M06-2X level, which also supports the assignment to a higher bond order in the complex. For comparison, three extra model molecules of FBFe(CO)$_3$, F$_2$BFe(CO)$_3^-$, and F$_2$BFe(CO)$_4^-$ complexes with triple, double, and single B–Fe bonds were also computationally designed. Their corresponding B–Fe bond distances as well as the relaxed force constants which have been successfully applied to quantify bond strength[44,45] are listed in Supplementary Table 2. The B–Fe bond length (1.61 Å) and relaxed force constant (not scaled, 481 N m$^{-1}$) of BFe(CO)$_3^-$ are shorter and larger than those of FBFe(CO)$_3$ (1.81 Å, 220 N m$^{-1}$), which involves a typical B≡Fe triple bonding. The bond length and relaxed force constant for double-bonded F$_2$B=Fe(CO)$_3^-$ (1.94 Å, 158 N m$^{-1}$) and single-bonded F$_2$B–Fe(CO)$_4^-$ (2.02 Å, 140 N m$^{-1}$) are in line with values for double and single bonds, respectively.

Figure 3 displays the canonical Kohn-Sham valence B–Fe nonbonding (11e), bonding (13a$_1$, 10e, 14a$_1$) and the corresponding antibonding orbitals (17a$_1$, 12e, 15a$_1$) of BFe(CO)$_3^-$. Their atomic orbital contributions are listed in Supplementary Tables 2 and 3. The highest occupied doubly degenerate 11e MOs are largely B–Fe non-bonding in character, and are primarily Fe 3d$_\delta$ atomic orbitals that comprise notable Fe-to-CO 2$\pi^*$ back-donation interaction. The 14a$_1$ MO is a $\sigma$-type B–Fe bonding orbital, which is composed of 28% Fe 3d$_z^2$ + 9% Fe 4p$_z$ and 26% B 2p$_z$ + 23% B 2s. The doubly degenerate 10e MOs are $\pi$-type B–Fe bonding orbitals, which are formed by the interactions between Fe 3d$_{xz/yz}$ (68%) and B 2p$_{x/y}$ (10%). Below these MOs, the 13a$_1$ MO composed of 45% Fe 3d$_z^2$ + 3% Fe 4p$_z$ + 3% Fe 4s and 40% B

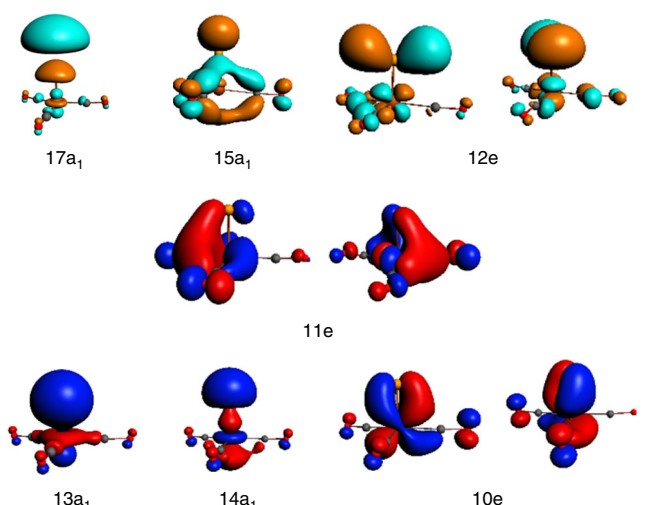

**Fig. 3** The canonical Kohn-Sham valence molecular orbitals of BFe(CO)$_3^-$. The B–Fe nonbonding (11e), bonding (13a$_1$, 10e, 14a$_1$) and the corresponding antibonding molecular orbitals (17a$_1$, 12e, 15a$_1$) are plotted with isosurfaces = 0.05 au

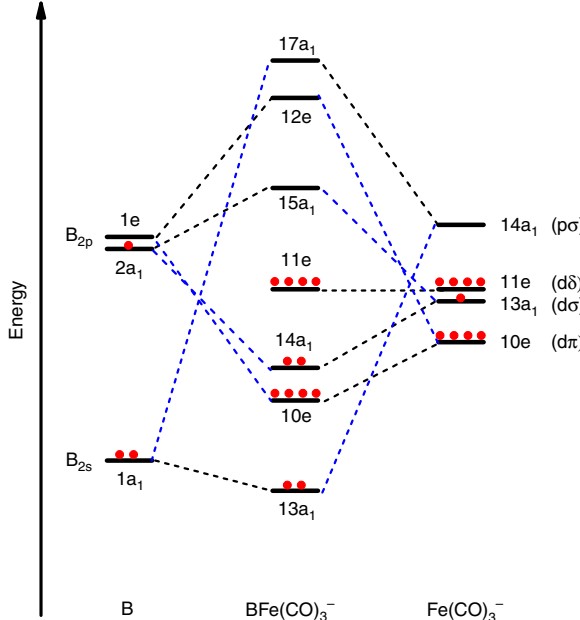

**Fig. 4** Bonding scheme of the C$_{3v}$ structure of $^1$A$_1$–BFe(CO)$_3^-$. The scheme qualitatively illustrates the bonding interactions between boron 2s-2p orbitals and Fe 3d/4p orbitals in the Fe(CO)$_3^-$ fragment

2s + 2% B 2p$_z$ is another σ-type B–Fe bonding MO. The results clearly show that there are four occupied MOs in BFe(CO)$_3^-$, which can be identified as two σ (14a$_1$ and 13a$_1$, hereafter σ$_p$ and σ$_s$ respectively) and two π (10e) B–Fe bonding interactions. The B≡Fe quadruple bonding assignment is also supported by adaptive natural density partitioning (AdNDP)[46] analyses shown in Supplementary Table 2. The semi-localized results indicate that the B–Fe bonding interactions in BFe(CO)$_3^-$ involve two 5c-2e σ bonds and two 5c-2e π bonds. It should be mentioned that here there is some mixing of the B–Fe and Fe-CO bonding MOs, but the major components and shapes of MOs resemble those of the classical electron-pair bonding models. The above bonding analysis also indicates that the B–Fe bonding interactions have quite small effect on the Fe-to-CO 2π* back-donation interactions in the Fe(CO)$_3^-$ fragment. Charge analyses in Supplementary Table 4 also indicate that the negative charge is largely located on the Fe(CO)$_3$ moiety. The antisymmetric CO stretching frequency (1841 cm$^{-1}$) of BFe(CO)$_3^-$ is only slightly higher than that of free Fe(CO)$_3^-$ (1780 cm$^{-1}$), but is lower than those of neutral Fe(CO)$_4$ and Fe(CO)$_5$.

Figure 4 qualitatively illustrates the bonding picture of BFe(CO)$_3^-$ and its correlation to the interacting fragments B and Fe(CO)$_3^-$. The major interactions between B and Fe(CO)$_3^-$ involve one σ$_p$-type (14a$_1$) Fe–B electron-sharing bonding, two degenerate π-type (10e) Fe→B dative bonding, and one σ$_s$-type (13a$_1$) B→Fe dative bonding, leading to the aforementioned quadruple bonding. The ground state BFe(CO)$_3^-$ anion can thus be regarded as being formed via the interactions between the ground state $^2$P-B atom with the (2s)$^2$(2p$_z$)$^1$ configuration and the Fe(CO)$_3^-$ fragment in its $^2$A$_1$ electronic ground state with (a$_1$)$^1$(e)$^4$ (e)$^4$ configuration. The B≡Fe quadruple bonding interactions are further characterized using the principal interacting orbital (PIO) approach[47]. The calculated PIOs and PIMOs shown in Supplementary Figs. 8–11 clearly reveal the fourfold bonding interactions between B and Fe in the BFe(CO)$_3^-$ anion, which further support the above assignment.

The strengths of the above-mentioned pairwise orbital interactions can be quantitatively estimated by the EDA-NOCV method[48,49]. The numerical results calculated at the M06-2X / TZ2P level are listed in Supplementary Table 5. The EDA-NOCV calculations suggest that 43% of the attractive forces in BFe(CO)$_3^-$ are due to orbital (covalent) interactions. The breakdown

of the orbital interaction term $\Delta E_{orb}$ into individual orbital contributions reveals that the four bonding components contribute 42.8% (σ$_p$), 46.6% (two π), and 8.2% (σ$_s$) of the orbital bonding interactions. Although the fourth component (σ$_s$ B→Fe dative bonding interaction) is much weaker than the electron-sharing σ$_p$ and two Fe→B dative π bonding interactions, it has a calculated interaction energy of 18.4 kcal/mol, indicating that it is a non-negligible, albeit weak, bonding interaction. The deformation densities $\Delta\rho(\sigma)$ and $\Delta\rho(\pi)$ that are connected to the σ and π interactions in BFe(CO)$_3^-$ are displayed in Supplementary Table 5. The deformation densities $\Delta\rho$ show the direction of the charge flow and the orbitals that are involved, where the color code of the charge flow is red → blue. The results clearly indicate the electron-sharing bonding nature of the stronger σ$_p$ interaction and the dative bonding character of the two π and the weaker σ$_s$ interactions. The weaker σ$_s$ interaction is strongly polarized with obvious charge flow from B atom to Fe(CO)$_3^-$. In contrast, the two degenerate π interactions show reversed charge flow from Fe(CO)$_3^-$ to B.

To make certain that these single-configuration quantum chemical methods are reliable, the wavefunction of BFe(CO)$_3^-$ is further examined in a multi-configurational framework. Ab initio CASSCF calculation involving 12 valence electrons in a space of 12 MOs was performed at the M06-2X optimized geometry, and the Löwdin natural orbitals (NOs) were analyzed using the CASSCF density matrix. The six strongly occupied and six weakly occupied natural orbitals and their occupation numbers (NOONs) are displayed in Supplementary Fig. 12. There is no significant occupation number of B–Fe antibonding NOs, $1\sigma^{*0.05}\pi^{*0.18}2\sigma^{*0.11}$, indicating that the multi-reference feature of this molecule is not very large. The B≡Fe effective bond order (EBO)[9] based on the ground-state CASSCF wavefunctions is calculated as 3.7, which is rather close to four and further supports the B≡Fe quadruple bonding assignment.

In summary, the BFe(CO)$_3^-$ anion complex has been identified by a combined quantum chemical and experimental study. The anion is generated in the gas phase and is studied by mass-selected infrared photodissociation spectroscopy. The complex

has been characterized to have a cylindrical $BFe(CO)_3^-$ structure with $C_{3v}$ symmetry and a very short B–Fe bond distance. Electronic structure and chemical bonding analyses indicate that the complex exhibits an unprecedented B≡Fe quadruple bonding interactions, featuring one electron-sharing $\sigma$ bond, two Fe→B dative $\pi$ bonds as well as one additional weak B→Fe dative $\sigma$ bonding interaction. The results extend the maximum bond order of boron element to four. This study reveals that a variety of such quadruple-bonded systems of main-group and transition-metal elements may exist in inorganic and organometallic chemistry.

## Methods

**Experimental details.** The anion complexes were generated in the gas phase using a pulsed laser vaporization/supersonic expansion ion source. A bulk target compressed from an isotopically enriched ($^{11}B$- or $^{10}B$-depleted) or natural abundance boron powder was used. The ions were produced from the laser vaporization process in expansions of helium seeded with 5−10% CO using a pulsed valve (General Valve, Series 9) at 0.5−1.0 MPa backing pressure. After free expansion and cooling, the anions were skimmed into a second chamber where they were pulse-extracted into a Wiley−McLaren type time-of-flight mass spectrometer. The anions of interest were mass-selected and decelerated into the extraction region of a second collinear time-of-flight mass spectrometer, where they were dissociated by a tunable IR laser. The tunable IR laser used is generated by a KTP/KTA//AgGaSe2 optical parametric oscillator/amplifier system (OPO/OPA, Laser Vision) pumped by a Nd: YAG laser, producing about 0.5−1.0 mJ per pulse in the range of 1600 −2200 cm$^{-1}$. Resonant absorption leads to fragmentation of the anion complex. The infrared photodissociation spectrum is obtained by monitoring the yield of the fragment anion as a function of the dissociation IR laser wavelength and normalizing to parent anion signal.

**Computational methods.** Geometry optimization and vibrational frequency calculations of the most stable structure of $BFe(CO)_3^-$ were performed using the density functional theory (DFT) methods with the PBE[50,51], B3LYP[52,53], and M06-2X[54] functionals using the ADF 2016 program packages [ADF 2016.101, http://www.scm.com]. The Slater-type-orbital (STO) basis sets with the quality of triple-$\zeta$ plus two polarization functions (TZ2P)[55] were applied with the consideration of scalar-relativistic (SR) effects at the zero-order regular approximation (ZORA)[56]. In order to obtain more accurate B–Fe bond distance, a series of single-point calculations at different B–Fe distances were carried out using a domain-based local pair natural orbital coupled cluster method, DLPNO-CCSD(T)[57–59], with the def2-TZVP basis sets (11s6p2d1f)/[5s3p2d1f] for the B, C and O, and (17s11p7d1f)/[6s4p4d1f] for Fe[60]. The DLPNO-CCSD(T) calculations were performed with the ORCA program package[61].

The chemical bonding properties were analysed by employing several different methods including adaptive natural density partitioning (AdNDP)[46] and the principal interacting orbital (PIO)[47]. The def2-TZVP basis sets, (11s6p2d1f)/[5s3p2d1f] for the B, C, and O, and (17s11p7d1f)/[6s4p4d1f] for Fe in Gaussian 09 program[62], were applied in the AdNDP calculations. For determining the most stable structure of $BFe(CO)_3^-$, $BFe(CO)_4^-$, and $BFe(CO)_5^-$, B3LYP with Dunning's correlation consistent basis sets aug-cc-pVTZ[63], (10s5p2d1f)/[4s3p2d1f] for B, C, and O, and (20s16p8d2f1g)/[7s6p4d2f1g] for Fe[64] were applied with Gaussian 09 program. The PIO analyses were also carried out using the Gaussian 09 program with aug-cc-pVTZ basis sets.

The multi-configurational complete-active-space SCF (CASSCF)[65] calculations were performed using the MOLPRO 2012.1 program package to examine the description of electronic configurations based on the DFT-optimized geometries with the Dunning's correlation consistent basis sets aug-cc-pVTZ[63], (10s5p2d1f)/[4s3p2d1f] for B, C, and O, and (20s16p8d2f1g)/[7s6p4d2f1g] for Fe[64]. The active space used 12 valence electrons in 12 orbitals ($12_e, 12_o$), which includes six occupied orbitals and six unoccupied orbitals at the frontier region of the molecular orbitals.

## Data availability

The data that support the findings of this study are available within the article and the associated Supplementary Data. Any other data are available from the corresponding author upon request.

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

## Acknowledgements

This work was supported by National Natural Science Foundation of China (grant nos 21433005, 21688102, 21590792, and 91426302). The calculations were done using supercomputers at the Southern University of Science and Technology (SUSTech) and Tsinghua National Laboratory for Information Science and Technology.

## Author contributions

The experiments were performed at East China University of Technology by C.C., L.M. and M.L. The theoretical calculations were performed at Tsinghua University by J.W., Y.Z. and W.L. M.Z., H.H. and J.L. wrote the paper and supervised the experimental and theoretical parts. All authors discussed the results and commented on the manuscript at all stages.

## Competing interests

The authors declare no competing interests.
