## [Transparent Peer Review File · Nature Communications]

Reviewers' comments:

Reviewer #1 (Remarks to the Author):

The paper reports detection of [FeB(CO₃)]⁻ and its extensive theoretical analysis. The authors claim that this complex contains a quadruple Fe-B bond. I think that the observation is in principle interesting and suitable for this journal. However, I have several points that should be addressed prior to publication of this finding.

1. The IR spectra are not sufficiently addressed. Firstly, it would be of course ideal, if the authors could measure the stretching frequency of the Fe-B bond. I do understand that they do not have an equipment for it. Therefore, analysis of the carbonyl stretches must be done in a more detail. Why are the bands so broad? Is it due to a rotational envelop? Can it be reproduced by calculations? How do the IR spectra of related [Fe(CO₄)]⁻ and [FeB(CO₅)]⁻ look like? Are the C=O stretches sensitive to the bond order between iron and boron?
2. The authors state that all other isomers of [FeB(CO₃)]⁻ than the suggested isomer with the terminal Fe-B bond lie higher in energy and therefore they were not considered. All the isomers with the calculated IR spectra should be in the Supporting Information. Also, I did not find analysis of a possible spin states of iron. The relative energies for different spin-state complexes should be listed too.
3. CO ligands make strong pi back-bonding interactions and the CO stretching frequencies reported here are in the range that is typical for metal carbonyl complexes. In fact, [Fe(CO₄)] and [Fe(CO)₅] have both higher CO frequencies. How does it correlate with the results presented here? If the back-donation is dominant for the Fe-B bond, how can this also lead to the red shift observed in the CO vibrations?

Reviewer #2 (Remarks to the Author):

The authors describe the identification of a quadruple B-Fe bond in an anion of C₃ symmetry. The study relies on the use of both experiments and several theoretical tools at different levels of approximation.

Reading through the papers I noted some minors improvements that could be made:

- 1/ p:3 there is a typo in the sentence: "whic[h] can be assigned to the 11BFe(CO)₃ .. "
- 2/ The software used for the DFT calculations is missing in the Methods part.
- 3/ Is there a reason as why so many softwares were used? (at least Orca, Gaussian and Molpro). Is it due to some analysis tools being available in particular packages? One would generally tend to avoid using different packages.
- 4/ The frequency calculated are scaled by a factor to "correct" them. Has this factor been determined by the author ? In which case it would be nice to simply add the results of the calculations in the SI.
- 4b/ The authors should add in the main text whether the force constant obtained by the calculation are scaled as well or not.
- 5/ Figure 3 would be clearer if the MO would be bigger, it is hard to see the details as it is now. Also, the top orbital is cropped.
- 6/ In Fig3, the captions read: "illustrating qualitatively the bonding interaction ..." however, the actual interaction is not shown on the left hand side of the figure. One would expect the dash line to show which orbital of both fragment interact to form the molecular orbitals.
- 7/ In figure S3, the authors show the calculated energy along the B-Fe bond at the DLPNO-CCSD(T) level. It is not said how the minimum has been determined. Is it by a fit ? If yes, the authors should add the expression of the function.

All in all, I think the paper is interesting and shows how complex can a "simple" molecule be. The

findings of the authors about the quadruple nature of the B-Fe bond are supported by several theoretical results which complete each others.

The topological analysis (AdNDP and PIO) illustrate especially well the nature of the bond.

I would recommend the paper to be publish upon correction/clarification of the points I mentionned earlier.

Reviewer #3 (Remarks to the Author):

The manuscript of Zhou, Li and coworkers concerns their gas-phase generation of an iron complex with a terminal, naked boron atom as ligand. Such a ligand would be expected to have three bonds to the metal, however, through their calculations, the authors show that a significant fourth bonding interaction exists, namely a sigma-type donation. While I cannot expertly critique the theory methods, the experimental work appears well done and in line with common practises in this type of synthesis.

My main concern with the manuscript is that I do not think the molecule should be described explicitly as having a "quadruple bond". However, the authors can quite believably state that there is a fourth, much weaker, bonding interaction. Calling this a quadruple bond is somewhat inflammatory (just like that between carbon atoms, which are described in the text: "this quadruple bond assignment has been debated, which has stimulated hot discussion"). The reason for this is that calling it a quadruple bond suggests that all four bonds are somewhat equal in weight, which does not reflect reality in either this system or the one of Shaik, Rzepa, etc. In reality the fourth bond is somewhat unimportant. Thus I would change the writing significantly to reflect this, i.e. by calling it a triple bond with a minor fourth bonding interaction, or specifying that four bonding interactions exist, the fourth being relatively minor.

Other than this, the generation and confirmation of the complex is quite an achievement, and the determination of the fourth bond is also important knowledge for this type of compound. Thus I am in favor of publication in principle.

A few other minor points:

- I would suggest changing the word "prepared" in the abstract to "generated" as the former implies that the complex has been made in quantity.

- The source of the iron in the complex is unclear from the text. Is it correct that the Fe comes from an impurity in the CO (except in the reaction with the labelled CO)? If so, it should be explicitly stated that the Fe in the sample comes from this impurity.

- Change "cylinder" to "cylindrical" on p7

- The detected anion $[\text{BFe}(\text{CO})_5]^-$ is not really discussed. A comparison of the bonding between these two complexes might be interesting. Also, there appears to be a MS signal between the tri and pentacarbonyl, presumably $[\text{BFe}(\text{CO})_4]^-$. Is this also relevant for discussion?

Responses to the Referees' Comments

We appreciate the valuable comments from the referees, which have significantly helped us to improve the quality of our manuscript. The revisions and responses are outlined as follows:

Reviewer #1:

1. The IR spectra are not sufficiently addressed. Firstly, it would be of course ideal, if the authors could measure the stretching frequency of the Fe-B bond. I do understand that they do not have an equipment for it. Therefore, analysis of the carbonyl stretches must be done in a more detail. Why are the bands so broad? Is it due to a rotational envelop? Can it be reproduced by calculations? How do the IR spectra of related $[\text{Fe}(\text{CO}_4)]^-$ and $[\text{FeB}(\text{CO}_5)]^-$ look like? Are the C=O stretches sensitive to the bond order between iron and boron?

Response: The Fe-B stretching frequency of $\text{BFe}(\text{CO})_3^-$ was predicted at the B3LYP level to absorb at 863 cm^{-1} with very low IR intensity (5 km mol^{-1}). Detection of such low frequency mode with such low IR intensity is not possible for us. This statement is now added in the manuscript (See the first paragraph of Page 5, second yellow highlighted sentences).

Following the reviewer's suggestion, we add more detailed discussions about the carbonyl stretching modes. A sentence is added to point out that "The bands are quite broad due to power broadening, as the dissociation is a multiphoton process." (See Page 4, the first yellow highlighted sentence).

In addition to the $\text{BFe}(\text{CO})_3^-$ anion, the infrared spectra of $\text{Fe}(\text{CO})_4^-$, $\text{BFe}(\text{CO})_4^-$ and $\text{BFe}(\text{CO})_5^-$ are measured and are shown in Supplementary Figure 3. The IR spectrum of $\text{BFe}(\text{CO})_3^-$ is quite different to those of $\text{Fe}(\text{CO})_4^-$, $\text{BFe}(\text{CO})_4^-$ and $\text{BFe}(\text{CO})_5^-$. These results are added in the manuscript (See Page 4, middle yellow highlighted sentences).

The simulated IR spectra of different $\text{BFe}(\text{CO})_3^-$ isomers in the carbonyl stretching frequency region are added as Supplementary Figure 4. The isomers with one and two CO ligands bonded to the boron center are predicted to be higher in energy than the most stable structure, with their spectra being quite different to that of the most stable $\text{BFe}(\text{CO})_3^-$ isomer. The OCB- $\text{Fe}(\text{CO})_2^-$ isomer is predicted to have a B-Fe bond length of 1.97 \AA , corresponds to a B-Fe single bond. The B-Fe bond distance of the $(\text{OC})_2\text{B-FeCO}$ isomer is predicted to be 2.15 \AA . This bond is characterized to be a B-Fe weak single bond. We point out that "The calculated infrared spectra of the three lowest-lying isomers are compared with the experimental spectrum in Supplementary Figure 4. Obviously, only the calculated spectrum of the most stable isomer matches the experiment." (See Page 4, last yellow highlighted sentences).

2. The authors state that all other isomers of $[\text{FeB}(\text{CO})_3]^-$ than the suggested isomer

with the terminal Fe-B bond lie higher in energy and therefore they were not considered. All the isomers with the calculated IR spectra should be in the Supporting Information. Also, I did not find analysis of a possible spin states of iron. The relative energies for different spin-state complexes should be listed too.

Response: The simulated IR spectra of different $\text{BFe}(\text{CO})_3^-$ isomers in the carbonyl stretching frequency region are added as Supplementary Figure 4. For each isomer, all possible spin states are calculated and the results are added in Supplementary Table 1.

3. CO ligands make strong pi back-bonding interactions and the CO stretching frequencies reported here are in the range that is typical for metal carbonyl complexes. In fact, $[\text{Fe}(\text{CO})_4]$ and $[\text{Fe}(\text{CO})_5]$ have both higher CO frequencies. How does it correlate with the results presented here? If the back-donation is dominant for the Fe-B bond, how can this also lead to the red shift observed in the CO vibrations?

Response: As we mentioned in the manuscript, the highest occupied doubly degenerate 11e MOs of $\text{BFe}(\text{CO})_3^-$ are largely B-Fe non-bonding in character, and are primarily Fe 3d atomic orbitals that comprise notable Fe-to-CO $2\pi^*$ back-donation interactions. These Fe-to-CO $2\pi^*$ back-donation interactions are responsible for the large red-shift of the carbonyl stretching frequencies. The MOs of the $\text{Fe}(\text{CO})_3^-$ fragment that involve in forming Fe→B dative bonds are its 10 e orbitals, which are mainly Fe 3d in character that comprise only weak Fe-to-CO $2\pi^*$ back donation interactions. Therefore, the B-Fe bonding has only small effect on the Fe-to-CO $2\pi^*$ back donation interactions in the $\text{Fe}(\text{CO})_3^-$ fragment. The antisymmetric CO stretching frequency (1841 cm^{-1}) of $\text{BFe}(\text{CO})_3^-$ is only slightly higher than that of free $\text{Fe}(\text{CO})_3^-$, which is observed at 1780 cm^{-1} in the gas phase, but is lower than those of neutral $\text{Fe}(\text{CO})_4$ and $\text{Fe}(\text{CO})_5$. Several sentences are added in the manuscript: “The above bonding analysis also indicates that the B-Fe bonding interactions have quite small effect on the Fe-to-CO $2\pi^*$ back donation interactions in the $\text{Fe}(\text{CO})_3^-$ fragment. The antisymmetric CO stretching frequency (1841 cm^{-1}) of $\text{BFe}(\text{CO})_3^-$ is only slightly higher than that of free $\text{Fe}(\text{CO})_3^-$ (1780 cm^{-1}), but is lower than those of neutral $\text{Fe}(\text{CO})_4$ and $\text{Fe}(\text{CO})_5$.” (See Page 6 first paragraph, yellow highlighted sentences).

Reviewer #2 (Remarks to the Author):

1. p:3 there is a typo in the sentence: "whic[h] can be assigned to the $11\text{BFe}(\text{CO})_3^-$.. "

Response: The typo error is corrected.

2. The software used for the DFT calculations is missing in the Methods part.

Response: The software used for the DFT calculations is now added in the Methods part (See Page 8 second paragraph, yellow highlighted sentences).

3. Is there a reason as why so many softwares were used? (at least Orca, Gaussian and Molpro). Is it due to some analysis tools being available in particular packages? One would generally tend to avoid using different packages.

Response: We do intend to avoid using different packages, but some analysis tools are only available or more professional in particular packages. For example, the

DLPNO-CCSD(T) calculation is only available in ORCA, while ADF is better for EDA-NOCV analysis. Molpro is much more efficient for CASSCF, CCSD(T) and CASPT2 calculations, especially for geometry optimizations.

4. The frequency calculated are scaled by a factor to "correct" them. Has this factor been determined by the author ? In which case it would be nice to simply add the results of the calculations in the SI. The authors should add in the main text whether the force constant obtained by the calculation are scaled as well or not.

Response: We mentioned in the footnote of Table 1 that the scale factors are taken from the ratio of the experimental frequency (2143 cm^{-1}) and the calculated harmonic frequency for free CO. The results are now added in Supplementary Table 6. The force constants are not scaled as now mentioned in the manuscript (Page 5, second paragraph).

5. Figure 3 would be clearer if the MO would be bigger, it is hard to see the details as it is now. Also, the top orbital is cropped.

Response: The MO contours have been shown in Figure 3 with clearer and bigger version, as well as the top orbital uncropped. The left part has been moved out and shown as a separated Figure (Figure 4).

6. In Fig3, the captions read: "illustrating qualitatively the bonding interaction ..." however, the actual interaction is not shown on the left hand side of the figure. One would expect the dash line to show which orbital of both fragment interact to form the molecular orbitals.

Response: The figure has been updated with added interaction on the left hand side as shown in Figure 4.

7. In figure S3, the authors show the calculated energy along the B-Fe bond at the DLPNO-CCSD(T) level. It is not said how the minimum has been determined. Is it by a fit ? If yes, the authors should add the expression of the function.

Response: Yes, it is fitted based on 7 points and the expression of the fitting function has been added in Supplementary Figure 5.

Reviewer #3:

1. My main concern with the manuscript is that I do not think the molecule should be described explicitly as having a "quadruple bond". However, the authors can quite believably state that there is a fourth, much weaker, bonding interaction. Calling this a quadruple bond is somewhat inflammatory (just like that between carbon atoms, which are described in the text: "this quadruple bond assignment has been debated, which has stimulated hot discussion"). The reason for this is that calling it a quadruple bond suggests that all four bonds are somewhat equal in weight, which does not reflect reality in either this system or the one of Shaik, Rzepa, etc. In reality the fourth bond is somewhat unimportant. Thus I would change the writing significantly to reflect this, i.e. by calling it a triple bond with a minor fourth bonding interaction, or

specifying that four bonding interactions exist, the fourth being relatively minor.

Response: We fully understand the referee's concern on whether the molecule should be described explicitly as having a "quadruple bond". In the literature, the first quadruple bond was reported by Cotton and coworkers in 1964 (F. A. Cotton et al. Science, 1964, 145, 1305). The Re-Re bond in Re_2Cl_8^- was assigned to be a quadruple bond. Besides one σ and two degenerate π bonds, there is an additional weak δ bond. Theoretical analyses (See for example: G. Frenking et al. Theor. Chem. Acc, 2008, 120, 313) indicate that the δ bond is in fact extremely weak. The EDA-NOCV analysis shows that the contribution of the δ bond to the total Re-Re bonding is only about 0.2%, which is much smaller than that of the σ bond (39.3%) and the two degenerate π bonds (29.9% each). In the case of the B-Fe bond in $\text{BFe}(\text{CO})_3^-$, the EDA-NOCV results (Supplementary Table 5) show that the fourth dative σ bond is weaker than the electron sharing σ and two dative π bonds, but it provides considerable contribution of 8.2% to the total B-Fe bonding. Following reviewer's suggesting, we changed the description of "quadruple bond" to "quadruple bonding interactions" in the abstract and conclusion section.

2. I would suggest changing the word "prepared" in the abstract to "generated" as the former implies that the complex has been made in quantity.

Response: The word "prepared" is changed to "generated".

3. The source of the iron in the complex is unclear from the text. Is it correct that the Fe comes from an impurity in the CO (except in the reaction with the labelled CO)? If so, it should be explicitly stated that the Fe in the sample comes from this impurity.

Response: We already mentioned in the manuscript that the iron carbonyl complexes come from trace of iron carbonyl impurity in the carbon monoxide sample.

4. Change "cylinder" to "cylindrical" on p7

Response: "cylindrical" has been changed to "cylindrical".

5. The detected anion $[\text{BFe}(\text{CO})_5]^-$ is not really discussed. A comparison of the bonding between these two complexes might be interesting. Also, there appears to be a MS signal between the tri and pentacarbonyl, presumably $[\text{BFe}(\text{CO})_4]^-$. Is this also relevant for discussion?

Response: The infrared spectra of $\text{BFe}(\text{CO})_4^-$ and $\text{BFe}(\text{CO})_5^-$ are also measured and are added as Supplementary Figure 3. The most possible structures based on primary DFT/B3LYP calculations are also shown in the Figure. As can be seen, the IR spectra of $\text{BFe}(\text{CO})_4^-$ and $\text{BFe}(\text{CO})_5^-$ are quite different to that of $\text{BFe}(\text{CO})_3^-$, implying different structure and bonding situation in these anion complexes. The predicted B-Fe distances in $\text{BFe}(\text{CO})_4^-$ (2.01 Å) and $\text{BFe}(\text{CO})_5^-$ (1.94 Å) are significantly longer than that of $\text{BFe}(\text{CO})_3^-$ (1.63 Å). These results are now mentioned in the manuscript. (Page 4, middle yellow highlighted sentences).

Reviewers' comments:

Reviewer #1 (Remarks to the Author):

I think that the binding in the presented $\text{BFe}(\text{CO})_3^-$ anion is of interest to the readership of this journal. I don't think that experimental data provide solid arguments to describe the structure of this anion (see the explanation below). However, anions with the given mass were undoubtedly detected and the suggested structure with a multiple bond between Fe and B is the most stable one according to DFT calculations. Hence, it is conceivable that the authors detected the anions that they describe in detail by theoretical calculations.

Specific comments to the interpretation of the IRMPD spectra:

I don't find the assignment of the IRMPD spectra convincing. It may be that the authors detected the declared isomer $\text{BFe}(\text{CO})_3^-$ (Suppl. Fig. 4a), but it also may be that they detected $(\text{CO})_2\text{BFe}(\text{CO})^-$ (Suppl. Fig. 4c) or a mixture of both. The energy difference between these two isomers is only 2 kcal/mol. Detection of $(\text{CO})_2\text{BFe}(\text{CO})^-$ would explain the broadening of the main experimental band towards lower wavenumbers. Further, IRMPD experiments are not linear in intensities of the detected bands. In fact, a cut-off in detection of low intense bands is expected unless ions with very low bond dissociation energies are investigated (for explanation see e.g. JPCA, 2019, 123, 4149, Fig. 2). Hence, I do not agree with the sentence: "Obviously, only the calculated spectrum of the most stable isomer matches the experiment".

Further, it would be instructive to show the DFT spectra along with the IRMPD spectra of the anions shown in Supplementary Figure 3. It would allow us to judge the level of agreement between DFT and IRMPD for this type of compounds.

Responses to the Referee's Comments

We appreciate the valuable comments from the referee, which helped us to further improve the quality of our manuscript. The revisions and responses are outlined as follows:

(1) I don't find the assignment of the IRMPD spectra convincing. It may be that the authors detected the declared isomer $\text{BFe}(\text{CO})_3^-$ (Suppl. Fig. 4a), but it also may be that they detected $(\text{CO})_2\text{BFe}(\text{CO})^-$ (Suppl. Fig. 4c) or a mixture of both. The energy difference between these two isomers is only 2 kcal/mol. Detection of $(\text{CO})_2\text{BFe}(\text{CO})^-$ would explain the broadening of the main experimental band towards lower wavenumbers. Further, IRMPD experiments are not linear in intensities of the detected bands. In fact, a cut-off in detection of low intense bands is expected unless ions with very low bond dissociation energies are investigated (for explanation see e.g. JPCA, 2019, 123, 4149, Fig. 2). Hence, I do not agree with the sentence: "Obviously, only the calculated spectrum of the most stable isomer matches the experiment".

Reply: We understand the referee's concern that the energy difference between the $\text{BFe}(\text{CO})_3^-$ structure (Suppl. Fig. 4a) and the $(\text{OC})_2\text{BFe}(\text{CO})^-$ isomer (Suppl. Fig. 4c) is very small and that both isomers may be detected. Based on a comparison between the simulated and experimental IR spectra shown in Suppl. Figure 4, the simulated spectrum of the C_{3v} $\text{BFe}(\text{CO})_3^-$ structure is most consistent with the experimental spectrum. The detection of $(\text{CO})_2\text{BFe}(\text{CO})^-$ cannot be fully ruled out but is less likely. The full width at half-maximum (fwhm) of the main experimental band is about 35 cm^{-1} , which is quite smaller than the peak separation between the lowest and highest bands of the $(\text{CO})_2\text{BFe}(\text{CO})^-$ isomer (about 78 cm^{-1}) predicted at the B3LYP level. The observed broad band towards lower wavenumbers is likely due to power broadening and the involvement of anions containing residual internal energy from the formation process. Following the referee's suggestion, we modified the statements as "The simulated spectrum of the C_{3v} $\text{BFe}(\text{CO})_3^-$ structure is most consistent with the experimental spectrum. But the detection of the low-lying $(\text{CO})_2\text{BFe}(\text{CO})^-$ isomer cannot be fully ruled out as the main experimental band is quite broad with a full width at half-maximum of about 35 cm^{-1} ." (See Page 5, yellow highlighted sentences).

The referee mentioned that a cut-off in detection of low intense bands is expected unless ions with very low bond dissociation energies are investigated. The effect was explained to be due to collisional deactivation during the IR multiple photon-excitation process, which reduces the IRMPD efficiency for IR transitions with a small absorption cross section. We would like to point out that the collisional deactivation effect does not exist in our experiments as the IR photodissociation takes place in high vacuum conditions without collision (about 10^{-5} Pa). In addition, the IR oscillator strength is predicted to be quite high for the $\text{BFe}(\text{CO})_3^-$ species. The symmetric and antisymmetric CO stretching modes of the $\text{BFe}(\text{CO})_3^-$ anion are predicted to have IR intensities of 294 and 3484 km/mol , respectively.

(2) Further, it would be instructive to show the DFT spectra along with the IRMPD spectra of the anions shown in Supplementary Figure 3. It would allow us to judge the level of agreement between DFT and IRMPD for this type of compounds.

Reply: The calculated spectra of the lowest-lying structures of $\text{Fe}(\text{CO})_4^-$, $\text{BFe}(\text{CO})_4^-$ and $\text{BFe}(\text{CO})_5^-$ along with the experimental IRMPD spectra are shown in Supplementary Figure 3A-C. The results show that the simulated spectrum of the second lowest energy structure of $\text{BFe}(\text{CO})_4^-$ (Suppl. Fig. 3B(b)) is most consistent with the experimental spectrum. This structure is predicted to be 5.9 kcal/mol higher in energy than the most stable structure at the B3LYP level. It seems that the simulated spectrum of structure (e) (Suppl. Fig. 3C(e)) matches the experimental spectrum of $\text{BFe}(\text{CO})_5^-$, suggesting that the experimentally observed $\text{BFe}(\text{CO})_5^-$ anion has a C_{3v} OCB- $\text{Fe}(\text{CO})_4^-$ structure, which is predicted to be 14.8 kcal/mol higher in energy than the most stable structure. These results imply that a high-lying isomer rather than the lowest energy structure is experimentally observed for both $\text{BFe}(\text{CO})_4^-$ and $\text{BFe}(\text{CO})_5^-$. Note that the observed structures of both $\text{BFe}(\text{CO})_4^-$ and $\text{BFe}(\text{CO})_5^-$ involve a $\text{Fe}(\text{CO})_4$ moiety. Experimentally, the $\text{Fe}(\text{CO})_4^-$ anion is always observed to be the most intense peak in the mass spectrum. Although the formation mechanism and dynamic is not clear, we assume that the $\text{BFe}(\text{CO})_n^-$ anions ($n=3-5$) are formed via the reactions of iron carbonyls $\text{Fe}(\text{CO})_3^-$ and $\text{Fe}(\text{CO})_4^-$ with boron. The anions produced by laser ablation are quickly cooled via supersonic expansion, which prevent further rearrangement reactions to form the most stable isomers. These results are now added in Supplementary Information (Supplementary Figure 3A-C and notes).

REVIEWERS' COMMENTS:

Reviewer #1 (Remarks to the Author):

With this report, I will repeat my previous statements and concerns.

The paper is of interest, because of the claimed four-fold bonding between iron and boron in the $[\text{FeB}(\text{CO})_3]^-$ anion. This is a purely theoretical construct and should be thus understood and evaluated as such.

The authors claim that $[\text{FeB}(\text{CO})_3]^-$ exists, because they can detect it by mass spectrometry. They prove the structure of these anions by IRMPD spectroscopy. Unfortunately, they couldn't examine the Fe-B bonding experimentally. Instead, they examined the C=O stretching range and assigned the spectra based on DFT calculations.

I do not find the assignment of the experimental IRMPD spectra based on the theoretical IR spectra convincing. The agreement between the IRMPD spectra of different ions ($[\text{Fe}(\text{CO})_4]^-$, $[\text{FeB}(\text{CO})_4]^-$, $[\text{FeB}(\text{CO})_5]^-$, $[\text{FeB}(\text{CO})_3]^-$) and theory is not good. For example, I do not find that any of the theoretical spectra of possible isomers of $[\text{FeB}(\text{CO})_4]^-$ agree with the experiment (Suppl. Figure 3B). This might suggest that the B3LYP method does not describe the structures or the IR spectra of these anions well. Also, it might be likely that authors form mixtures of ions with various structures, especially, if they freeze-out the initially formed isomers as they suggest in the Supplementary Information.

Disregarding the IRMPD experiments, the authors detected anions with the mass corresponding to the $[\text{FeB}(\text{CO})_3]^-$ composition and it is likely that at least part of these anions has the structure described in this paper. Hence, this experimental evidence can be used as a starting point for the main part of the paper which is the theoretical part.

Response to the Referee's Comments

We are grateful to the referee 1 for the questions and criticism that have helped us to clarify and improve the manuscript. We are especially thankful to him/her for pointing out the possibility of co-existence of the various isomers of $\text{BFe}(\text{CO})_3^-$ anion. While DFT methods at the different levels of PBE, B3LYP, M06-2x are usually enough to judge the relative energies, in our case of the $\text{BFe}(\text{CO})_3^-$ anion it surprisingly is not the case as revealed by the highly accurate CCSD(T) data now available (see below). Without the referee's question we would probably have missed this critical point.

Reviewer #1 (Remarks to the Author):

Questions: With this report, I will repeat my previous statements and concerns.

The paper is of interest, because of the claimed four-fold bonding between iron and boron in the $[\text{FeB}(\text{CO})_3]^-$ anion. This is a purely theoretical construct and should be thus understood and evaluated as such.

Response: We thank the referee for recognizing the novelty of the four-fold bonding interactions between iron and boron in the $\text{BFe}(\text{CO})_3^-$ anion. Given the referee's comments, we also made effort in changing "quadruple bonds" to "quadruple bonding interaction" in the whole manuscript. We have altered sentences to avoid viewing experiment as a proof for Fe-B bonding. The computational study and experimental evidence are presented in a way more like that the experiment supports the theoretical predictions.

Questions: The authors claim that $[\text{FeB}(\text{CO})_3]^-$ exists, because they can detect it by mass spectrometry. They prove the structure of these anions by IRMPD spectroscopy. Unfortunately, they couldn't examine the Fe-B bonding experimentally. Instead, they examined the C=O stretching range and assigned the spectra based on DFT calculations.

I do not find the assignment of the experimental IRMPD spectra based on the theoretical IR spectra convincing. The agreement between the IRMPD spectra of different ions ($[\text{Fe}(\text{CO})_4]^-$, $[\text{FeB}(\text{CO})_4]^-$, $[\text{FeB}(\text{CO})_5]^-$, $[\text{FeB}(\text{CO})_3]^-$) and theory is not good. For example, I do not find that any of the theoretical spectra of possible isomers of $[\text{FeB}(\text{CO})_4]^-$ agree with the experiment (Suppl. Figure 3B). This might suggest that the B3LYP method does not describe the structures or the IR spectra of these anions well. Also, it might be likely that authors form mixtures of ions with various structures, especially, if they freeze-out the initially formed isomers as they suggest in the Supplementary Information.

Response: The referee is absolutely right that it is indeed unfortunate that the B-Fe vibrational frequency is too low and too weak to be detected experimentally. We added this sentence: “the B-Fe stretching frequency of $\text{BFe}(\text{CO})_3^-$ cannot be directly observed as it was predicted to absorb at 863 cm^{-1} with very low IR intensity (5 km mol^{-1}) at the B3LYP level.”

As for the assignment of the experimental IRMPD spectra based on the theoretical IR spectra, the B3LYP functional indeed does not describe the energies of these anions well. In trying to answer the referees’ questions, we have now performed very high level ab initio calculations using DLPNO-CCSD(T) method, in addition to the original DFT calculations at the levels of PBE, B3LYP, and M06-2X. We have now found that at the high level of DLPNO-CCSD(T) method, the other two possible isomers of $\text{BFe}(\text{CO})_3^-$ lie 27.8 and 42.5 kcal/mol higher in energy than the quadruple-bonded C_{3v} $\text{BFe}(\text{CO})_3^-$ structure. This result essentially precludes the co-existence of the two high-lying isomers with the most stable quadruple-bonded C_{3v} $\text{BFe}(\text{CO})_3^-$ structure because CCSD(T) is the quantum chemistry “golden standard” and its error bar is no more than 3-5 kcal/mol in general (see *Nature Communications*, volume 10, Article number: 2903, 2019). Even though the B3LYP energies are not accurate, their structure and vibrational frequencies are usually acceptable, as is known in theoretical chemistry. As shown in Supplementary Fig. 6, the experimental spectrum of C_{3v} $\text{BFe}(\text{CO})_3^-$ matches the B3LYP simulated spectrum well, except that the experimental spectrum is quite broad, which is likely due to power broadening and/or the involvement of hot anions.

Regarding on the $\text{BFe}(\text{CO})_4^-$ and $\text{BFe}(\text{CO})_5^-$ anions, it is clear that none of the simulated single spectrum of $\text{BFe}(\text{CO})_4^-$ isomers agrees with the experimental spectrum. The observed $\text{BFe}(\text{CO})_4^-$ anion is likely a mixture involving the three lowest-lying isomers predicted at the DLPNO-CCSD(T) level (Supplementary Fig. 4). As shown in Supplementary Fig. 5, the experimentally observed $\text{BFe}(\text{CO})_5^-$ anion is mainly due to the most stable C_{3v} $(\text{OC})\text{B}-\text{Fe}(\text{CO})_4^-$ structure, likely with minor contribution from the second and third lowest-lying structures as predicted at the DLPNO-CCSD(T) level.

Questions: Disregarding the IRMPD experiments, the authors detected anions with the mass corresponding to the $[\text{FeB}(\text{CO})_3]^-$ composition and it is likely that at least part of these anions has the structure described in this paper. Hence, this experimental evidence can be used as a starting point for the main part of the paper which is the theoretical part.

Response: The referee’s doubt on our experimental IRMPD spectra assignment of the C_{3v} $\text{BFe}(\text{CO})_3^-$ structure lies on whether the other isomers can co-exist and therefore contribute to the broad IR spectra. As just mentioned, the issue is now solved, to our point of view, with the DLPNO-CCSD(T) results. It is very unlikely that the other isomers of $\text{BFe}(\text{CO})_3^-$, which are predicted to lie 27.8 and 42.5 kcal/mol higher in energy than the quadruple-bonded C_{3v} $\text{BFe}(\text{CO})_3^-$ structure can co-exist in our

experimental condition. As suggested by the referee, we have altered sentences to avoid viewing experiment as a proof for Fe-B bonding. The computational study and experimental evidence are presented in a way more like that the experiment supports the theoretical predictions.